# Molecular Insights into the Flavivirus Replication Complex

**DOI:** 10.3390/v13060956

**Published:** 2021-05-21

**Authors:** Kaïn van den Elsen, Jun Ping Quek, Dahai Luo

**Affiliations:** 1Lee Kong Chian School of Medicine, Nanyang Technological University, Singapore 636921, Singapore; 2NTU Institute of Structural Biology, Nanyang Technological University, Singapore 636921, Singapore; 3Living Systems Institute, University of Exeter, Exeter EX4 4QD, UK; 4School of Biological Sciences, Nanyang Technological University, Singapore 637551, Singapore

**Keywords:** flavivirus, replication complex, non-structural protein, RNA replication

## Abstract

Flaviviruses are vector-borne RNA viruses, many of which are clinically relevant human viral pathogens, such as dengue, Zika, Japanese encephalitis, West Nile and yellow fever viruses. Millions of people are infected with these viruses around the world each year. Vaccines are only available for some members of this large virus family, and there are no effective antiviral drugs to treat flavivirus infections. The unmet need for vaccines and therapies against these flaviviral infections drives research towards a better understanding of the epidemiology, biology and immunology of flaviviruses. In this review, we discuss the basic biology of the flavivirus replication process and focus on the molecular aspects of viral genome replication. Within the virus-induced intracellular membranous compartments, flaviviral RNA genome replication takes place, starting from viral poly protein expression and processing to the assembly of the virus RNA replication complex, followed by the delivery of the progeny viral RNA to the viral particle assembly sites. We attempt to update the latest understanding of the key molecular events during this process and highlight knowledge gaps for future studies.

## 1. Introduction

One large clade of known viruses is the positive-sense single-stranded RNA (+ssRNA) virus class, which belongs to group IV in the Baltimore classification [1]. These viruses include many important human pathogens, such as rhinoviruses, coronaviruses and flaviviruses. The genome of these viruses has the same polarity as the messenger RNA of the host cell, allowing it to take advantage of cellular mechanisms and be readily translated into protein [2]. This triggers a mass restructuring of the cellular landscape. These viral proteins are associated with cellular membranes, manipulating their structure to generate specialised organelle-like structures for the replication of viral RNA [3]. Cellular signalling and trafficking are subverted, recruiting host factors to the structures and shielding them from host surveillance systems [4].

Flavivirus is a genus of +ssRNA viruses that includes many clinically serious human pathogens, including dengue virus, West Nile virus, Zika virus, Japanese encephalitis virus, yellow fever virus and tick-borne encephalitis. These viruses are typically transmitted to humans through arthropods (such as mosquitoes or ticks), and thus, are classified as arboviruses [5]. When combined, these viruses account for significant mortality and morbidity across the globe. Dengue, for instance, infects approximately 390 million people per year, of which 500,000 experience severe disease and 25,000 die [6,7]. Additionally, the number of dengue cases reported to the WHO has undergone an 8-fold increase over the past two decades: from approximately half a million in 2000 to over 4 million in 2019 [8]. Currently, there is no effective vaccine or antiviral drug to treat dengue virus, despite continuing efforts to develop one. During infection, these viruses induce the formation of mini-organelles associated with the endoplasmic reticulum, which act as viral RNA replication factories. Within these replication organelles (ROs), multiple viral and host factors combine to form the RNA replicase machinery, also known as replication complex (RC). These proteins perform crucial functions during viral replication, making them enticing drug targets. The key to treating these diseases may well lie in producing vaccines and antivirals aimed at the proteins in the replication complex and their molecular mechanisms. In this study, we provide a summary of the current understanding of this highly specialised structure. We focus on the molecular structures of the nonstructural proteins and their interactions with each other and RNA, as well as the ultrastructural characterisation of the complex by electron microscopy and tomography. 

## 2. The Flavivirus Replication Cycle

Flavivirus virions enter a susceptible cell through the process of receptor-mediated endocytosis. Once the virion is internalised, it is released from the receptor, and its membrane fuses with the endosomal membrane, releasing the ribonucleocapsid into the cytoplasm [9]. The nucleocapsid is degraded, uncoating the viral RNA, which is then translated by ribosomes on the rough endoplasmic reticulum (RER) (Figure 1A). This yields a viral polyprotein containing structural and nonstructural proteins that are anchored to the ER membrane [10]. The polyprotein is then cleaved by the viral protease NS3 and host proteases into its constituent proteins. The nonstructural proteins assemble into the replication complex and drive the invagination of the ER membrane to produce replication organelles [11]. The replication complexes replicate the viral RNA through a negative strand RNA intermediate, yielding a positive strand RNA that is then packaged into new nucleocapsids and envelopes, creating immature virions (Figure 1B) [12]. These immature virions are secreted in vesicles and enter the Golgi apparatus, where they progress through chambers of decreasing pH [13]. The enzyme furin cleaves the envelope proteins of the virus, producing the mature virions that then undergo exocytosis (Figure 1A) [14].

## 3. Function and Formation of the Replication Organelle

### 3.1. Function

The key function of the replication organelle is to replicate the viral RNA. The replication organelle achieves this replication by producing a complementary negative strand RNA that acts as an intermediate for the generation of many new + strand viral genomes. Additionally, the RO acts to facilitate the packaging of this RNA into new virions. The multiple biosynthetic steps involved in this process require a close association of the various necessary factors. The compartmentalisation of the process in this way ensures the spatiotemporal synchrony of these steps, dramatically improving their efficiency and speed [22]. Furthermore, the compartments provide an area shielded from the hostile host cellular milieu. Host nucleases and proteases are quick to degrade any foreign matter, and if cytoplasmic sensors detect viral RNA, an innate immune response will be triggered to detect and destroy viral matter. Therefore, the membrane delimitation of these complexes is a highly advantageous and evolutionarily conserved property of flaviviruses and other +ssRNA viruses [23].

### 3.2. Biogenesis

Once the +ssRNA genome of flaviviruses enters the cell, it is translated into a large single polyprotein by the host ribosomes on the endoplasmic reticulum (ER). This contains both the structural proteins that form the capsid (C, prM and E) and the nonstructural proteins (NS1, NS2a, NS2b, NS3 NS4a, NS4b and NS5) [24]. These nonstructural proteins perform the intracellular functions of the virus. After the polyprotein is generated, it is inserted into the membrane of the adjacent ER and is cleaved into its constituent parts by the coactivity of the viral protease NS3 and the host proteases. The nonstructural proteins proceed to induce curvature in the ER membranes, forming spherical invaginations into the ER lumen with a pore connecting them to the cytoplasm. Here, NS proteins assemble into viral replication complexes, in conjunction with the host proteins and the viral RNA [25]. Alongside these well-defined complexes, the formation of morphologically distinct and far less organised membranous structures, known as convoluted membranes (CMs), is also induced.

#### 3.2.1. Replication Organelle Curvature

NS4a has been identified in some studies as the key driver of these membrane alterations, inserting itself into the ER membrane and undergoing oligomerisation to construct a scaffold onto which the replicase complex assembles. Membrane topological studies of NS4a reveal a luminal helix that lies parallel to and closely associates with the ER membrane, flanked by transmembrane helices. This topology may function as a wedge that, when polymerised, induces the cytosolic facing curvature seen in the RCs [26]. Membrane invaginations can be formed with NS4a alone, but they differ in morphology from wild-type RCs, suggesting other nonstructural proteins, such as NS4b, are involved in their biogenesis [27,28]. Conversely, a more recent study has found that NS1 also has the capability of remodelling the ER membrane to create RO-like structures. Yali Ci et al. identified that NS1 dimers insert three hydrophobic regions into the ER membrane on the luminal surface, thereby attaching to the membrane. This mechanism creates a negative curvature that creates dips in the ER surface that progress to form spherical invaginations. NS1 can form these structures alone, which results in structures much more similar to ROs than those produced by NS4a [29]. Further compelling evidence has suggested that the 3′ stem-loop region of the dengue virus genome (3′SL) has a major involvement in the generation of RO curvature. A study by Cerikan et al. found that the polyprotein can not induce the formation of vesicle packets (VPs) alone, but their biogenesis can be rescued by the addition of this 3′ terminal untranslated region (UTR) [30]. It is possible that all three of these elements play a role in the establishment of these structures and/or their stability, and more research will need to be carried out to understand this process entirely.

#### 3.2.2. Host Factors in Assembly

Flaviviruses depend upon many host factors to perform a diverse array of functions in RC formation. One key group in this coalition is host lipids, which are essential in many stages of the flaviviral life cycle. A crucial lipid in membrane biogenesis is fatty acid synthase, which is recruited and co-opted by flaviviruses to produce lipids that are integrated into the membrane to provide curvature and fluidity [31]. Flavivirus ROs are contingent upon the presence of cholesterol. It is a component of all mammalian cellular membranes and the ROs are particularly enriched with it, presumably to aid in biosynthesis and improve fluidity [32]. Other lipid types that flaviviruses have evolved to utilise are sphingolipids and ceramides. They have several functions, including the regulation of the cell cycle and homeostasis of the cell. Ceramides are known to induce negative membrane curvature and analyses of dengue-induced cell rearrangements have implicated sphingolipids in their formation [33,34]. RO biogenesis is also dependent on the cooperation of many host proteins. Although several interacting host proteins have been identified, their roles in the replication cycle are not well characterised. One of the better understood interacting host proteins is reticulon-3. This membrane protein interacts with NS4a and localises to membrane regions with high curvature. Reticulon-3 is likely also partially responsible for the membrane shaping [35]. Finally, the HSP40 family protein DNAJC14 has been demonstrated to have numerous roles in the regulation of flaviviral replication, including membrane shaping. Although DNAJC14 does not have any membrane curving mechanisms of its own, it has been implicated in producing curvature indirectly. A proposed model for this phenomenon is through the targeting of the replication proteins, the establishment of interactions and attachment to membrane microdomains. This induces conformational changes in the NSPs and alters the characteristics of the membrane, thereby orchestrating RC formation. [36,37]. Recently, the ER-resident host proteins TMEM41B and VMP1 have also been implicated in RO formation through a CRISPR knockout screen. Flaviviruses may hijack the membrane manipulation capabilities these proteins use for autophagosome formation and corrupt them to drive RO formation. Intriguingly, single nucleotide polymorphisms in TMEM41B that hinder flaviviral replication are present in over 20% of the population of South East Asia, a region with a high prevalence of flavivirus infections [38]. The establishment of these replication factories through membrane curvature is a crucial step in the flavivirus replication cycle. Although many possible mechanisms for this step have been suggested, there are some contradictions between them, preventing us from identifying a definitive model. More research needs to be conducted on this process and how it may be inhibited, as a better understanding of this step could lead to promising new therapeutic avenues.

## 4. Architecture of the Replication Organelle

In recent years, electron microscopic, tomographic and immuno-EM analysis studies have greatly enhanced our understanding of the flaviviral replication complex and the network of membranes in which it resides. Once the altered cellular membranes of infected cells were identified, many subsequent studies were conducted to unravel their mysteries. Early EM research allowed their discovery and categorisation into VPs and CMs. Successive studies have localised viral proteins within these networks, indicating that the VPs correspond to the site of viral RNA replication and that CMs contain the site of polyprotein processing.

### 4.1. Structure, Composition and Arrangement

A study by Welsch et al. was key in developing our understanding [24]. Here, these authors built on previous research on the composition of the ROs, placing NS1, NS2b, NS3, NS4A, NS4b and NS5 inside the ROs. dsRNA, an intermediate of RNA replication, was shown to co-localise with the NS proteins, solidifying this location as the site of viral RNA replication (Figure 2). They went on to further characterise their structure and arrangement in detail using electron microscopy and tomography. This revealed spherical invaginations into the ER lumen consisting of a single layer of ER-derived membrane. Tomographic analysis highlighted channel-like openings of 10 nm connecting the structures to the cytoplasm [24]. This channel surfaces close to the ribosomes of the rough ER where the newly produced RNA is translated into protein. The pore putatively provides an entrance for cytoplasmic factors requisite for replication, including nucleotides. Additionally, it provides the means of egress for newly synthesised strands of + RNA [39]. Directly opposing the pores is the site of the viral budding events, further demonstrating the importance of efficient spatial orchestration of the events in the viral life cycle. These findings paint a picture of an interconnected network of membranes that tightly couples the different stages of replication [24].

### 4.2. Convoluted Membranes

Convoluted membranes, on the other hand, consist of a network of much less structured membranes in smooth ER bundles associated with ROs and mitochondria (Figure 1A). The function of these is not well established. They do not contain viral RNA and are therefore unlikely to participate in RNA replication [24]. They are, however, enriched with viral proteins, chiefly NS3 and NS2b. Since these are both associated with polyprotein cleavage, the CMs are presumed to be the site of polyprotein processing and maturation. CMs may also be a location for the storage of proteins or lipids. A recent study by Chatel-Chaix et al. linked NS4b in convoluted membranes to the perturbation of mitochondrial morphology, elongating them and depleting the mitochondria-associated ER membranes. This could hamper the innate immune response of the cells by obstructing interferon activation, thereby creating a cellular milieu favourable for viral replication [40]. Interestingly, CMs do not occur in mosquito cells infected by dengue or in neural progenitor cells infected by Zika, while ROs occur in both. This casts doubt upon the notion that they perform a key role in replication as their function is cell-type specific [41,42]. These inconsistencies may reflect alternate strategies used by the virus to adapt to distinct host cells. Overall, these membranes are an interesting element in the viral manipulation of host cellular dynamics and further study is required to elucidate how the interplay between ROs and the host influences viral propagation.

### 4.3. Host Factors in RC Function

In addition to the host factors required for RO assembly, flaviviruses hijack many other host factors that they rely on to reconfigure the infected cell. These factors are used to rewire cellular signalling pathways and traffic proteins and organelles to create a favourable environment for replication. Typically, these factors interact directly with a viral protein or the vRNA and their functions are often similar to those regularly used to support the cell, but are subverted for viral purposes. Recently, several studies have identified key host factors involved in flavivirus replication by combining affinity purification of viral proteins or RNA with mass spectrometry (MS) of the resulting isolates. This can be combined with small interfering RNA (siRNA) screening or CRISPR-Cas9 knockout production to identify the possible functions of these proteins in replication.

One study using affinity purification MS in combination with RNAi screening to produce a virus–host protein–protein interaction map made several interesting discoveries. The authors found that the chromatin-associated complex PAF1C, which is key in transcriptional elongation, interacted with NS5 in all four DENV serotypes, as well as West Nile virus (WNV) and ZIKV. Through this interaction, NS5 inhibits the recruitment of this transcription complex to interferon-stimulated genes, thereby inhibiting their expression and diminishing the host immune response. Additionally, they found that SEC61, a protein involved in ER function and insertion of transmembrane proteins into the ER, interacted highly specifically with NS4a. They demonstrated that the pharmacological modulation of this translocon inhibited DENV and ZIKV replication in human and mosquito cells. Finally, they identified an interaction between NS4a of ZIKV and ANKLE2, a gene previously associated with microcephaly, and showed that inhibition of ANKLE2 by NS4b likely contributes to the microcephaly induced by ZIKV [43].

RNA-centric approaches have also identified several flaviviral host factors involved in replication. Protein–viral RNA interactions are at the core of the molecular pathogenesis of flaviviruses and are essential for translation, replication and packaging. Mass spectrometry of proteins that bind to the viral RNA is therefore a powerful approach to identify host proteins involved in flaviviral replication. This approach has identified ER-localised ribosome-binding proteins (RBPs) as being implicated in replication. One such protein, RRBP1, which promotes ribosome-independent association of mRNAs with the ER, is likely co-opted to stimulate translation of the viral polyprotein. Furthermore, the RBP vigilin, a protein that binds to ribosomal RNA to promote mRNA translation, was also highly enriched in the identified interactome. While RRBPI activity is most pronounced in the early stages of infection, that of vigilin is more prominent in the later stages. Taken together, these results provide a basis for the host involvement in viral RNA translation and indicate their distinct roles [44]. Using a UV cross-linking approach combined with MS to study RBPs that bind DENV RNA has also yielded interesting results, which were then further investigated through siRNA gene silencing. Notably, the RBPs YBX1, PABPC1 and CSDE1, among others, were found to interact with the vRNA. They determined that YBX1 likely acts in the late stages of replication, possibly in genome packaging or viral egress. PABPC1 and CDSE1 also bind the vRNA, and may be involved in its translation, synthesis or stability [45]. These strategies are robust and effective in the purification of ribonucleoprotein complexes and their identification and play a key role in elucidating the relationships of viruses with their host cells.

CRISPR-Cas9 genetic screens have also recently come into the fore as a method to study flaviviral pathogenesis. These have enormous potential to both recognise and validate host factors involved in infection, as they can identify candidate genes without bias, and reliably knockout the gene to study their impact. Astonishingly, in studies where enriched host factors identified in CRISPR screes were knocked out, viral replication was shown to decrease 100–10,000-fold. These studies have identified many proteins that are hijacked in flaviviral infections. These include ER proteins such as ERAD, which is an ER protein that normally acts to target misfolded proteins for degradation [46]. Similarly, SEC61A1 and SEC63 form the translocon-associated protein complex in the ER and promote the translocation of proteins into the ER during translation [47]. Furthermore, STT3A and STTA3, components in the OST complex that typically catalyses glycosylation of proteins, have been shown to be a crucial factor in viral RNA synthesis. The OST complex can bind several nonstructural proteins, and therefore may function as a scaffold and act to synchronise RC assembly [46].

These studies have given us great insights into the complex and interconnected network of interactions between the host and the virus and demonstrate the scale of the overhaul the virus performs on cellular systems. Flaviviruses and hosts have coevolved for millennia and have developed strategies over time to co-opt a multitude of cellular proteins to aid in viral replication, becoming reliant on them. Therefore, the concept of targeting host factors to arrest viral replication has garnered support and could provide a higher resistance barrier and even protect against infections from multiple different viruses. However, this is not necessarily the case. Although replication can be impeded by targeting a host factor the virus depends on, the virus may evolve to circumvent its reliance on these factors. This is the case for the host factor TMEM41B, which induces curvature in RO formation. Mutations in NS4a and NS4b allowed ZIKV and YFV to continue replicating in TMEM41B KO cells [38]. High resistance barriers could be achieved through the targeting of host proteins that the virus relies on in both human and arthropod cells, such as the aforementioned SEC61.

## 5. Nonstructural Proteins

The viral genome is translated into a single polyprotein that will be cleaved by both host and viral proteases into three structural proteins: capsid (C), precursor membrane (prM) and envelope (E) and seven nonstructural (NS) proteins: NS1, NS2A, NS2B, NS3, NS4A, NS4B and NS5. The viral polyprotein is represented in Figure 3.

In recent decades, biochemical and functional studies of the nonstructural proteins have been well characterised. The following sections will only focus on highlighting the roles of the nonstructural proteins in the formation and functioning of the replication complex.

### 5.1. NS1

NS1 is a glycosylated protein with a molecular weight of approximately 48 kDa. It exists as a homodimer after post-translational modification in the ER lumen. Upon secretion into the extracellular space, it exists as hexameric lipoprotein particles. Within each NS1 monomer is contained twelve conserved cysteines that form six pairs of disulphide bonds and two N-linked glycosylation sites at residues N130 and N207, with a third glycosylation site at N175 for DENV and WNV [48,49]. A detailed description of the structure of NS1 can be found in the articles written by various groups [17,48,49]. The protein structures of NS1 published in the Protein Data Bank are also summarised in Table 1.

Both the dengue and Kunjin virus NS1 proteins were also observed to co-localise with the dsRNA [11,50]. Deletion or mutations of numerous flavivirus NS1 proteins abolished viral RNA replication and accumulation [29,51,52,53]. Further studies have revealed that NS1 involvement in viral RNA replication can be due to its role in ER modelling and the formation of vesicle packets [29,54]. NS1 was also proposed to regulate the formation and activity of the replication complex via the interaction with NS4A, NS4B and the NS4A–2K–NS4B intermediate [54,55,56,57].

**Table 1 viruses-13-00956-t001:** A summary of the protein structures of the NS1 proteins published in the Protein Data Bank.

Viruses	PDB IDs	Remarks	Space Group	Ref.
DENV-1	4OIG	NS1	C121	[18]
	6WEQ	NS1 in complex with neutralising 2B7 Fab	P12_1_1	[58]
DENV-2	4O6B	NS1	H3	[49]
	6WER	NS1 in complex with neutralising 2B7 Fab	I4_1_22	[58]
	7K93	NS1 in complex with neutralising 2B7 single chain Fab variable region (scFv)	P2_1_2_1_2_1_	
	7BSC	C-terminal NS1 in complex with 1G5.3 Fab	P2_1_22_1_	[59]
JEV	5O19	C-terminal NS1	I2_1_2_1_2_1_	[60]
WNV	4OIE	NS1	P3_1_21	[18]
	4OII	NS1 In complex with neutralising 22NS1 Fab	C121	
	4O6C	NS1	P3	[49]
	4O6D	NS1	P321	
YFV	5YXA	C-terminal NS1	P2_1_2_1_2_1_	[61]
ZIKV	5GS6	NS1 from 2015 Brazil strain	P4_1_22	[48]
	5IY3	NS1	C222_1_	[62]
	5K6K	NS1	I222	[17]
	5X8Y	NS1	C222_1_	[63]
	7BSD	C-terminal NS1 in complex with 1G5.3 Fab	P2_1_2_1_2_1_	[59]

### 5.2. NS2A

NS2A is a transmembrane protein with a molecular weight of approximately 22 kDa. The N-terminus of the protein is cleaved by an unknown host protease and localised on the ER lumen, while the C-terminus is found in the cytoplasm and cleaved by viral NS2B-3 protease.

Similar to NS1, Kunjin NS2A was shown to be co-localised with the double-stranded RNA in the VPs [64]. Mutagenesis studies demonstrated dengue and Zika NS2A to be involved in both RNA replication and virion assembly [65,66]. However, the trans-complementary assay only recovers virion production activity and not RNA replication, suggesting that two different sets of NS2A at different cellular locations are involved in each process [65,67]. The role of NS2A in the virus assembly process was recently reviewed by Barnard et al., 2021 [68]. However, more studies are required to fully understand the role of NS2A in viral RNA replication.

### 5.3. NS2B

NS2B is made up of 130 amino acid residues and has a molecular weight of approximately 14 kDa. It consists of a central hydrophilic region at the cytoplasmic side of the ER, that is flanked by two transmembrane helices at both the N and C termini of the proteins [69,70,71]. The membrane-associated residues of NS2B were also revealed by NMR spectroscopy studies [69,70]. The central hydrophilic region alone is adequate to act as the cofactor for the folding and enzymatic activity of the NS3 protease [71]. The NS2B cofactor was also shown to contribute to the substrate selectivity of the NS3 protease [72].

In addition to acting as the cofactor for NS3, NS2B also helps to anchor NS3 onto the ER membrane [73]. NS2B is also part of the replication complex and has extensive interaction with other membrane-bound NS proteins, serving a central role in bringing the other components of the complex together [24,74]. A study by Li et al. revealed that the correct topology of a4 helix in NS2B is essential for NS2A–NS2B interaction and that disrupting this interaction can impact virion assembly [75]. However, the exact role of the transmembrane regions of NS2B in virion assembly remains to be elucidated.

### 5.4. NS3

NS3 is a 68 kDa multifunctional protein. It consists of two main domains, an N-terminal protease and a C-terminal helicase. Together with the NS2B cofactor, the NS3 N-terminal protease forms a functional viral protease essential for the processing of the viral polyproteins. The protease cleaves the viral polyprotein at the capsid, NS2A/2B, NS2B/3, NS3/4A, NS4A/2K and NS4B/5. The viral protease adopts a chymotrypsin-like fold with two β-barrels and has a catalytic triad (H51, D75, S135) located at the cleft between these two β-barrels [76,77]. The N-terminal cytoplasmic residues (49 to 67) of NS2B help to stabilise the folding of the NS3 protease through the insertion of a β-sheet into the N-terminal β-barrel [76,77]. The C-terminal residues (68 to 98) of NS2B help to form the catalytic site [71,76,77,78,79].

The NS3 C-terminal helicase belongs to the superfamily 2 (SF2) helicases. The helicase is essential for unwinding the duplex RNA into single-stranded RNA for NS5 polymerase activity. It also possesses NTPase and RTPase activity. The helicase is composed of three domains. Domains 1 and 2 have α/β RecA-like folds, which is a characteristic of superfamily 1 (SF1) and SF2 helicases [80,81,82]. The RNA binding site is located in a tunnel that separates domains 1 and 2 from domain 3, while the ATP binding site is located between domains 1 and 2 [80,81,82]. A P-loop region also exists in domain 1, which is essential for NTP binding and catalysis.

Upon RNA binding, subdomain 3 rotates about 10° away from the RNA binding tunnel, enlarging the groove to accommodate the ssRNA [83,84]. Furthermore, RNA binding is also sufficient to allow inwards movement of the P-loop in the absence of ATP [83]. Despite the slight differences in the conformation of the P-loop of the various flavivirus helicases under apo conditions [80,81], upon RNA and ATP binding, they undergo different local rearrangements to adopt similar binding modes and structures [84,85].

Structural studies of the full-length NS3 protein have revealed that it adopts an elongated structure with a protease domain position near the NTP binding site of the helicase [16,86,87]. Two different conformations have been captured, with different relative orientations between the protease and helicase domains (Figure 4). Due to the flexibility of the interdomain linker region, the protease domain can rotate from about 160° to 180° relative to the helicase domain, depending on the flavivirus [86,87]. The length of the linker between the two domains has been shown to play a regulatory role [86]. The full-length protein structures of NS3 published in the Protein Data Bank are summarised in Table 2.

Most studies are in consensus and have revealed that the helicase domain has no effects on the protease activity [86,87,88]. However, there are conflicting findings with regard to the influence of proteases on the helicase domain. This could be due to intrinsic viral differences, protein construct design or methodology. Some studies revealed that in the absence of the protease, helicase unwinding activity for both dsRNA and dsDNA is decreased [16,82,89]. Other studies have suggested partial or no influence of the protease on the helicase domain [87,90]. Other findings include protease regulating and restricting NS3 helicase unwinding activity to double-stranded RNA [88].

A dimeric model for NS3 protein was also proposed [91]. The dimerisation may also be facilitated by the membrane association of full-length transmembrane NS2B, where it has been shown to dimerise in a cell-based assay [92].

**Figure 4 viruses-13-00956-f004:**
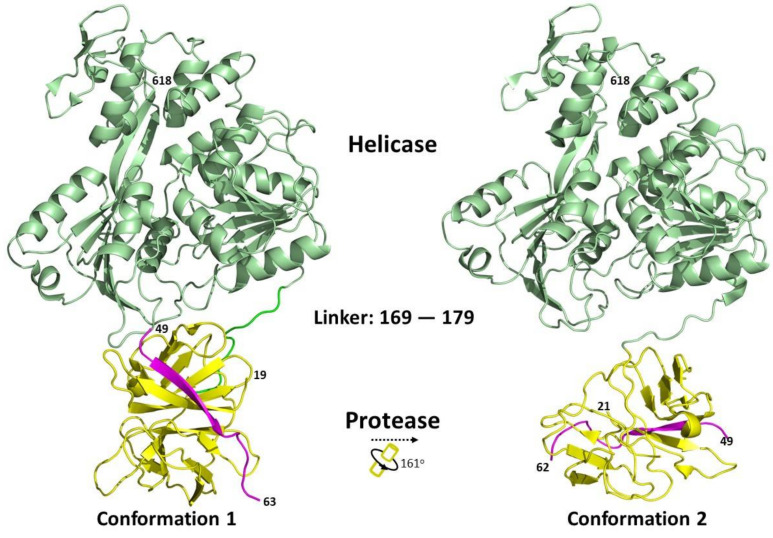
Relationship between different conformational states of the *Flavivirus* full-length NS3 structures. The protein structures were obtained from PDB and adapted from [16,86]. Conformation 1 (PDB: 2VBC) and conformation 2 (PDB: 2WHX) are shown. Ribbon representation showing the arrangement of the protease and helicase domains of NS3. The NS2B cofactor is coloured in magenta, protease domain is in yellow and the helicase domain in green. The conformational changes between the protein structures were analysed using the DynDom database [93].

**Table 2 viruses-13-00956-t002:** A summary of the full-length protein structures of the NS3 proteins published in the Protein Data Bank.

Viruses	PDB IDs	Protease Active Site	Remarks	Ref.
DENV-4	2VBC	Open	Partial NS2B cofactor linked to NS3 via a G_4_SG_4_ linker	[16]
	2WHX	Open	Partial NS2B cofactor linked to NS3 via a G_4_SG_4_ linker	[86]
	2WZQ	Open	Partial NS2B cofactor linked to NS3 via a G_4_SG_4_ linker, glycine insertion mutant between E173 and P174	
	5YVJ	Open	Partial NS2B cofactor linked to NS3 via a G_4_SG_4_ linker	[94]
	5YVU	Closed	NS2B cofactor linked to NS3 via a G_4_SG_4_ linker, in complex with BPTI	
	5YVV	Closed	NS2B cofactor linked to NS3 via a G_4_SG_4_ linker	
	5YVW	Closed	Unlinked NS2B cofactor	
	5YVY	Closed	Unlinked NS2B cofactor	
	5YW1	Closed	Unlinked NS2B cofactor, in complex with BPTI	
MVEV	2WV9	Open	Partial NS2B cofactor linked to NS3 via a G_4_SG_4_ linker	[87]

### 5.5. NS4A

NS4A is a membrane protein with three N-terminal amphipathic helices and four C-terminal membrane helices [26,95]. The last transmembrane helix, also known as the 2k peptide, acts as a signal sequence for the translocation of NS4B into the lumen of the ER. The 2k peptide is cleaved at the N-terminus by the NS2B-3 protease, while the C-terminus is cleaved by host signalase in the ER lumen.

Dengue and Kunjin NS4A have also been found to be co-localised with dsRNA, suggesting a role in viral RNA replication [26,64]. Studies have determined that NS4A is sufficient to induce cytoplasmic membrane rearrangement; DENV uses fully processed NS4A while KUNV requires full-length NS4A [26,27]. Dengue NS4A was also found to form oligomers mediated by the first transmembrane regions of NS4A. The mutation that disrupted the oligomerisation was found to abolish viral replication [96,97]. These findings suggest that NS4A oligomerise and act as a scaffold for the formation of VPs. In addition, WNV NS4A also regulates the ATPase activity of the NS3 helicase [98].

### 5.6. NS4B

NS4B is the largest membrane protein found in the flavivirus genome. It consists of nine helices with helices 2, 3, 5, 7 and 9 forming transmembrane domains 1–5, respectively [99,100,101]. DENV accumulation at ER-derived membranes is mediated by the transmembrane domains 3–5 [100]. Transmembrane domain 5 has also been proposed to undergo conformational changes and flip from the cytoplasmic side to the ER lumen upon NS4B–NS5 junction cleavage [99,100]. The cytoplasmic loop and transmembrane domain 4 and 5 were found to be responsible for NS4B dimerisation [102]. Recently, the ER membrane protein complex was also identified to act as an ER chaperone to facilitate the correct insertion and topology of the transmembrane region of NS4B [103,104,105].

NS4B has also been suggested to play a role in viral RNA replication as it is found localised with the other NS proteins that are involved in viral replication [100]. NS4B promotes the dissociation of ssRNA from NS3, enhancing the unwinding activity of the NS3 helicase domain [106].

### 5.7. NS5

NS5 protein carries two enzymatic domains, an N-terminal methyltransferase (MTase) domain and a C-terminal RNA-dependent RNA polymerase (RdRp). The MTase domain methylates the RNA sequentially to form the type I capped structure using S-adenosylmethionine (SAM) as the methyl donor [107]. The MTase was also proposed to have guanylyltransferase (GTase) activity, to transfer GTP to a 5′ diphosphorylated RNA [108]. RNA capping occurs when NS5 bind to GTP and form an NS5–GMP intermediate. Concurrently, NS3 RTPase will remove the γ-phosphate from the viral genome 5′ end, producing a diphosphorylated RNA intermediate product. NS5 will subsequently transfer the GMP to the diphosphorylated RNA upon RNA binding. Then, the RNA will reposition the guanosine near the SAM binding site to methylate the N7 atom of guanosine. The RNA will then undergo another repositioning to allow the initiating adenosine to be positioned near the SAM binding site for 2′O methylation. (ppA-RNA > GpppA-RNA > ^N7M^GpppA-RNA > ^N7M^GpppA_2′OM_-RNA). The RNA capping mechanism is illustrated in Figure 5. The RdRp domain is involved in RNA replication in both the de novo initiation and the elongation steps, helping to replicating the viral genome. However, the exact mechanism of RNA polymerase catalysis as well as the proposed conformational changes during the transition from the initiation to elongation phase remains to be elucidated.

The crystal structures of the MTase domain of various flaviviruses have been determined, which adopt a similar globular fold, that can be divided into three subdomains [109]. The first subdomain is characterised by a helix-turn-helix motif followed by a β-strand and an α-helix. The GTP-binding pocket can be found in this subdomain and was proposed to coordinate the GTP moiety of RNA substrate during 2′-O-ribose methylation [110,111]. The second subdomain is the core domain with a Rossmann fold, consisting of seven β-sheets and four α-helices. Both the active site for methyltransferase activity and the SAM and RNA binding sites can be found in the core domain. The conserved catalytic tetrad (KDKE) is also located near the SAM-binding site [112]. The last subdomain consists of an α-helix and two β-strands.

The crystal structure of the RdRp domain revealed a right-hand architecture consisting of a palm subdomain, a thumb subdomain and a fingers subdomain (Figure 6) [113,114,115,116,117]. The palm subdomain is the most structurally conserved region of NS5. The palm subdomain also houses the active site with two catalytic aspartic acid residues, required for chelating two Mg metal ions and the formation of the phosphodiester bond. The crystal structures also reveal a unique region known as the priming loop protruding from the thumb subdomain. It is a unique feature of RdRp and acts as a platform to catalyse de novo RNA initiation [113,114,115,116,117].

**Figure 5 viruses-13-00956-f005:**
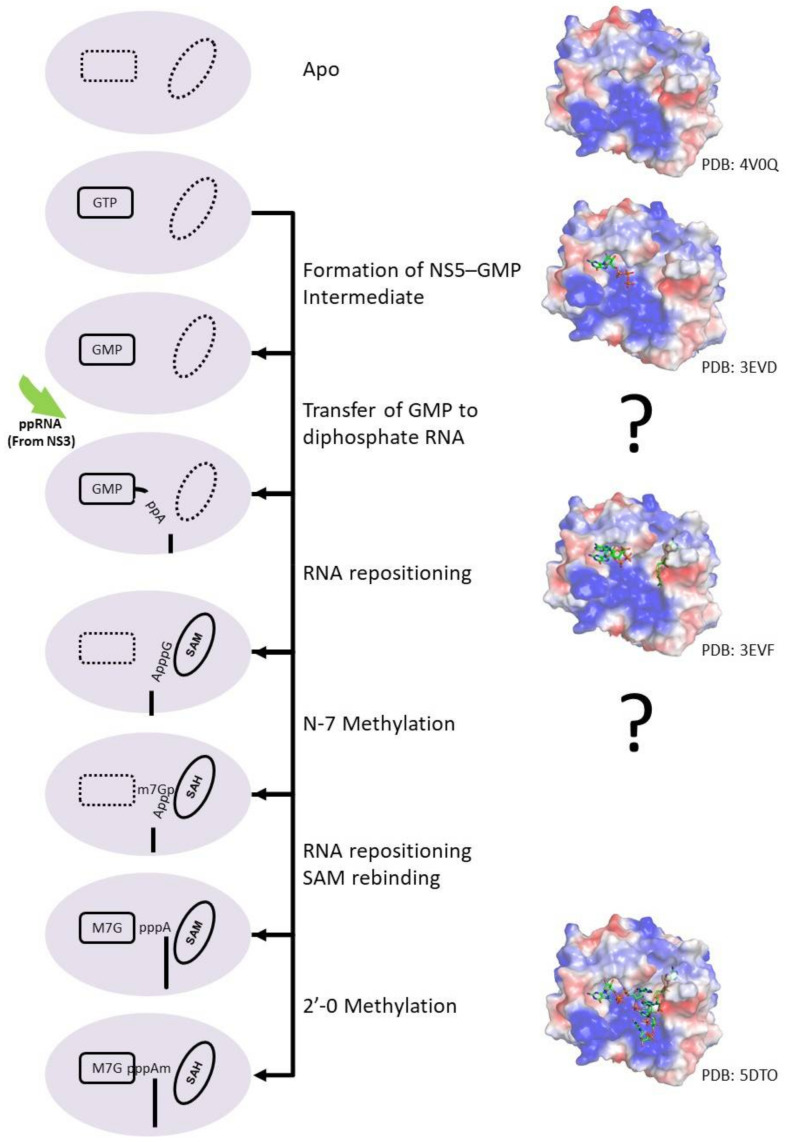
Cartoon representation of the flavivirus capping process. During the RNA capping process, GTP binds to the GTP binding pocket of NS5 and forms an NS5–GMP intermediate. Concurrently, NS3 RTPase removes phosphate from the viral genome to form a diphosphorylated RNA product. The NS5–GMP intermediate subsequently transfers the GMP to the diphosphorylated RNA upon RNA binding. Then, GpppA-RNA reposition the guanosine near the SAM binding site for N7-methylation. The SAH molecule is replenished by a SAM molecule while the ^N7M^GpppA-RNA undergoes another repositioning to allow the initiating adenosine to be positioned near the SAM binding site for 2′O methylation to generate ^N7M^GpppA_2′OM_-RNA. Representative protein structures with their PDB codes are also shown.

Numerous X-ray crystallography studies have revealed a series of different global conformations and different modes of interaction between the MTase domain and the RdRp domain for full-length NS5 proteins from different flaviviruses (Figure 6) [15,115,116,117,118,119]. In solution, NS5 from different DENV serotypes were also found to display multiple conformations [120]. The first conformation is observed in JEV and ZIKV [116,117,119]. It has a more elongated conformation with an interacting interface consisting of mostly hydrophobic residues. In DENV-3, the more compact conformation 5 is observed [115]. In this conformation, the MTase has an approximately 110° rotation to RdRp and establishes mainly polar contacts instead. Recently, three additional conformations were observed in YFV and DENV-2 [15,118,121]. Conformations 2 and 3 resemble the first conformation through rotation along an axis of the highly conserved GTR residues, which was previously proposed to pivot MTase movement relative to RdRp. The rotation leads to a partial opening of the MTase–RdRp interface. Conformation 4 resembles conformation 5 and is related by a 14° rotation along an axis near the interface and the MTase–RdRp linker region. The full-length protein structures of NS5 published in the Protein Data Bank are summarised in Table 3.

The ability of NS5 to adopt different global conformations was proposed to be mediated by the linker domain. In the crystal structure of full-length DENV-3, the linker adopts a short 3_10_ helix structure comprising of four residues [115]. Mutations to the linker region, disrupting the crosstalk between the MTase and the RdRp, are detrimental to virus replication [15,122].

**Table 3 viruses-13-00956-t003:** A summary of full-length protein structures of the NS5 proteins published on Protein Data Bank.

Viruses	PDB IDs	Ligands	Remarks	Space Group	Ref.
DENV-3	4V0Q	SAH and Zn	Conformation 5	P2_1_2_1_2	[115]
	4V0R	GTP, Mg, SAH and Zn			
	5DTO	M7G, Mg, SAH and Zn			[111]
	5CCV	SAH and Zn			[123]
	5JJR	Compound 29, SAH, Mg and Zn			[124]
	5JJS	Compound 27, SAH, Mg and Zn			
DENV-2	5ZQK	SAM, Mg and Zn	Conformation 3	P12_1_1	[15]
	6KR2	SAH and Zn	Conformation 3	P12_1_1	[118]
	6KR3	SAH and Zn	Conformation 4	C222_1_	
JEV	4K6M	SAH and Zn	Conformation 1	H3	[119]
YFV	6QSN	SAH and Zn	Conformation 2	P2_1_2_1_2_1_	[121]
ZIKV	5TFR	SAH and Zn	Conformation 1	P2_1_2_1_2	[125]
	5TMH			P22_1_2_1_	[116]
	5U0B			P2_1_2_1_2	[117]
	5M2X			P2_1_2_1_2_1_	[126]
	5M2Z			P6_5_	
	6I7P			P2_1_2_1_2_1_	

**Figure 6 viruses-13-00956-f006:**
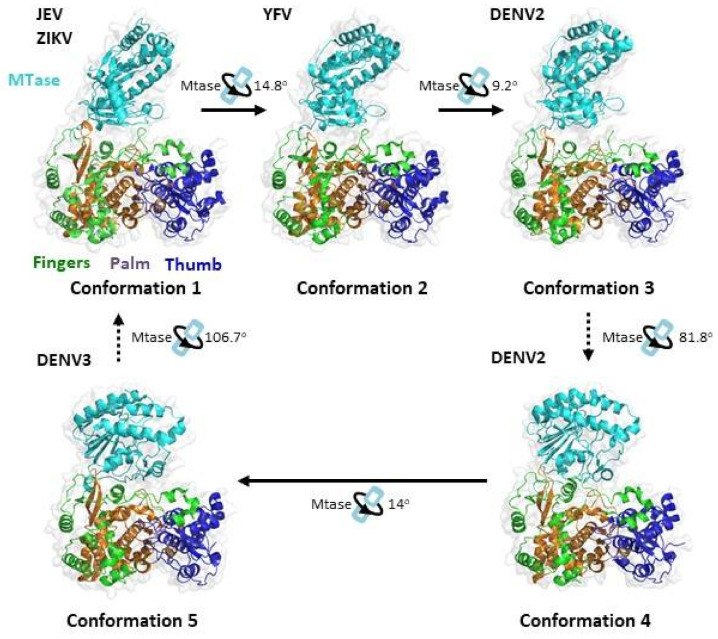
Relationship between the different conformational states of the *Flavivirus* full-length NS5 structures. The protein structures were obtained from PDB and adapted from [115,118,119,121]. JEV (PDB: 4K6M), YFV (PDB: 6QSN), DENV2 (PDB: 6KR2), DENV2 (PDB: 6KR3) and DENV3 (PDB: 4V0Q). Ribbon representations show the arrangement of the MT and RdRp domains of NS5. The MTase is coloured in cyan, the RdRp fingers in green, the RdRp palm in brown and the RdRp thumb in blue. The conformational changes between the protein structures were analysed using the DynDom database [93].

Biochemical studies suggest that the MTase domain plays different roles in the regulation of the polymerase activity of DENV RdRp [115,123,127,128]. These findings are also observed in JEV and Zika, where MTase might differentially regulate both the de novo initiation and elongation phases during viral RNA synthesis [118,129]. Rusanov et al. suggested that MTase enhances the stability of the NS5–RNA elongation complex [130]. The interaction between methyltransferase and the RdRp domain in Zika virus has been shown to be necessary for virus replication [15,117,122,130].

## 6. Interaction Network within the Replication Complex

Many studies have shown that all seven nonstructural proteins can be found in the replication complex [24,26,50,64,100,131]. NS3 and NS5 are the only two viral proteins that have enzymatic functions, thus they are proposed to form the core of the RC. The rest of the NS proteins are proposed to play regulatory roles to support viral replication. The interactions between the NS proteins are summarised in Table 4.

NS1, which resides in the ER lumen, was also proposed to transmit signals across the ER membrane to regulate viral RNA replication via the interaction with NS4A and NS4B for YFV and WNV, respectively [55,56]. However, the mechanism for signal transduction remains to be further studied. Recently, DENV NS1 (residues G161 and W168) was also found to interact with the NS4A–2K–NS4B intermediate [54]. Domain II of DENV NS1 was found to interact with residue Y41 of NS4A in the NS4A–2K–NS4B intermediate [57]. Further studies are required to understand the role of these interactions. The interaction was proposed to regulate NS4A–2K–NS4B cleavage for proper membrane association and maturation of NS4A and NS4B.

The transmembrane proteins have also been observed to interact with each other. The correct topology of the a4 helix in NS2B is essential for the NS2A–NS2B interaction [75]. NS4A (residues 40–76) and NS4B (residues 84–146) were also found to interact with one another. Furthermore, compensatory mutations were also found in NS4B when NS4A was mutated for both DENV and JEV, supporting the NS4A–NS4B interaction [132,133,134]. Li et al. conducted an interaction study using FRET and BiFC and revealed the additional interaction partners NS2A and NS4A, NS2B and NS4A and between NS2B and NS4B for WNV [74].

Extensive interactions have been found between the NS proteins for regulating NS3 enzymatic activity. First, the central hydrophilic region of NS2B interacts with the NS3 protease domain and acts as the cofactor for enzymatic activity and stability of the NS3 protease [71]. Second, WNV NS4A regulates the ATPase activity of the NS3 helicase [98]. Next, the cytoplasmic loop of NS4B (residue Q134) was found to interact with domains II and III of NS3 helicase [106,134,135]. This interaction allows the NS4B dimer to promote the dissociation of ssRNA from NS3 and enhance helicase unwinding activity [106]. Finally, NS3 helicase activity is also stimulated and upregulated by NS5 mediated by the NS3–NS5 interaction [89,136,137].

N570 of NS3 and K330 of NS5 have been shown to be the key residues involved in the NS3–NS5 interaction [138,139]. Further electrostatic potential analysis of NS3 (residues 566–585) and NS5 (residues 320–341) shows negative charges and positive charges, respectively, supporting the hypothesis that the two regions interact via charge interactions [139]. Optimal NS3–NS5 interaction affinities are proposed to be essential in coordinating RNA unwinding by NS3 helicase and RNA synthesis by NS5 RdRp [138,139].

**Table 4 viruses-13-00956-t004:** Interactions between the NS proteins within the replication complex.

Virus	Protein: Residues	Protein: Residues	References
DENV	NS1 (G161, W168)	NS4A-2K-NS4B	[54]
	NS1: 30–180	NS4A-2K-NS4B: 35–61 (Y41)	[57]
	NS2B	NS3	
	NS3: Helicase Domain II and III	NS4B: 129–165 (Q134)	[106,134,135]
	NS3: 566–585 (N570)	NS5: 320–341 (K330)	[138,139]
	NS4A: 40–76 (L48, T54, L60)	NS4B: 84–146	[140]
	NS4B	NS4B	[102]
WNV	NS1	NS4B	[55]
	NS2A	NS2B	[74]
	NS2A	NS4A	[74]
	NS2B	NS3	[74]
	NS2B	NS4A	[74]
	NS2B	NS4B	[74]
	NS3	NS5	[74]
	NS4A	NS4B	[74]
JEV	NS2A	NS2B	[75]
	NS3	NS5	[141]
YFV	NS1	NS4A	[56]

## 7. The Viral RNA

Flaviviruses have a single-stranded positive-sense RNA genome that is approximately 11,000 nucleotides long. The genome has a type 1 cap structure at its 5′ end with no poly(A) tail at its 3′ end. The cap structure is essential for protection against 5′-3′ exonucleases and cap-dependent translation of the flavivirus genome. The RNA genome is made up of a single open reading frame that is flanked by 5′ and 3′ untranslated regions (UTRs) (Figure 7) [142,143,144,145]. The following section will focus on RNA elements in the viral genome that are involved during RNA replication.

### 7.1. The 5′ UTR

The 5′UTR is approximately 100 nucleotides long, with different lengths for the different flaviviruses. The 5′ terminus consists of an N7MGpppA2′OM type I cap structure that is followed by a conserved AG dinucleotide [146,147]. The conservation of the dinucleotide sequence can be attributed to the ATP-specific priming site by the NS5 RNA-dependent RNA polymerase domain [148]. The 5′UTR has two conserved stem-loop domains, stem-loop (SL) A and SLB, which are separated by an oligo(U) spacer for the proper functioning of the two stem-loops. The SLA is around 70 nucleotides long and acts as a promoter for the NS5 protein to initiate RNA synthesis at the 3′ end of the cyclised genome [149]. The top and loop were found to be crucial for high-affinity binding of NS5 to the 5′UTR [149]. In addition to SLA, a highly conserved RNA duplex region termed 5′UAR-flanking stem (5′ UFS) has been identified and shown to interact with NS5 [150]. This structure is destabilised upon genome cyclisation, which is suggested to promote the translocation of NS5 from the 5′UTR to the 3′UTR for viral RNA synthesis [150]. Next, the SLB is around 30 nucleotides long and has 5′ upstream AUG region (UAR) sequences for long-range RNA–RNA interactions with the 3′UAR to form the cyclised RNA structure. Following SLB, 5′ downstream AUG region (DAR) sequences for long-range RNA–RNA interaction for genome cyclisation were also identified [151]. Other RNA elements within the 5′ ends include the 5′ cyclisation sequence (CS), which is located within the capsid protein-coding region, that interacts with the 3′CS for genome RNA circularisation. A capsid-coding region hairpin (cHP), which is found in the coding sequence and just downstream of the translation initiation codon AUG, also helps in starting codon recognition and RNA circularisation [152,153]. cHP was proposed to aid starting codon initiation by stalling the scanning 48S complex, enhancing translation of the viral polyproteins. This stalling mechanism was found to be dependent on both RNA structural stability and relative distance to the start codon, but not the sequence of the cHP [152].

### 7.2. The 3′UTR

The length of the 3′UTR varies for different flaviviruses and ranges from 400 to 600 nucleotides in length. It is divided into three domains. Domain 1 is located immediately after the stop codon and has the most variable nucleotide sequence within the 3′UTR. Although the functionality of the variable region is currently unknown, the variable region contributes to the formation of stem-loop (SL) structures, which helps to exert different functions in different hosts. Studies have also shown that these stem-loop structures are resistant to degradation to host exonuclease Xrn-1 and are referred to as xrRNA-1 and/or xrRNA-2 [154,155,156]. Next, domain 2 is moderately conserved with a tandemly duplicated dumbbell structure (DB1 and DB2), that contains the conserved sequences and repeated conserved sequence domains. The DB structures are further stabilised by the pseudoknot (PK), which plays a functional role during viral RNA synthesis, translation and subgenomic flaviviral RNA (sfRNA) formation [155,157]. Finally, domain 3 is highly conserved and is characterised by CS1, a small hairpin (sHP) and a large 3′SL. The 3′SL is functionally important as it interacts with host and viral proteins to modulate viral RNA synthesis and translation [142,143,144]. Recently, it was also reported that 3′SL contributes to vesicle packet formation [30].

### 7.3. Genome Cyclisation and Long-Range RNA–RNA Interactions

The flavivirus RNA genome exists in two conformations, linear and cyclised forms, and the balance between the two conformations is essential for viral replication (Figure 7) [158]. Genome cyclisation occurs via the base pairing of sets of inverted complementary sequences at the terminal ends of the flavivirus genome. The sets of inverted complementary sequences are 5′UAR-3′UAR, 5′DAR-3′DAR, 5′CS-3′CS and 5′ C1 of capsid region-3′ DB1 [144,151,159,160]. Numerous host chaperones have also been found to bind to the UTR regions of the viral genome and are proposed to be involved in genome cyclisation [161,162,163,164].

Through numerous functional studies of the elements in the flavivirus genome, genome cyclisation has been proposed to be essential for negative-sense viral RNA replication [144]. Genome cyclisation helps to bring the polymerase–promoter complex near the 3′ transcription initiation site and promotes the correct conformation of the 3′ end of the genome during initiation [144,159]. Furthermore, the cyclisation also destabilised the 5′ UFS, disrupting the interaction with NS5 and promoting the translocation of NS5 from the 5′UTR to the 3′UTR for viral RNA synthesis [150]. Genome cyclisation also inhibits de novo translation initiation of viral proteins via inhibition of ribosome scanning [165]. This finding suggests that flaviviruses use genome cyclisation as a molecular switch to regulate viral polyprotein translation and viral RNA replication, switching between linear translation-competent RNA forms to a circular replication-competent RNAs.

In addition to genome cyclisation, defined secondary and tertiary RNA structures mediated by long-range RNA–RNA interactions can also be found within the coding regions of the viral genome [166,167,168]. Different studies have identified different sets of secondary structures in the coding regions of the genome, from sixteen to twenty four for DENV and twelve to twenty four for ZIKV, due to the differences in methodology and analysis criteria [166,167,168]. However, all studies indicated that the RNA is more structured in virions than in the more extended forms in infected cells, suggesting that such structures could aid in genome packaging into virions. Depending on the lifecycle of the virus, RNA structural heterogeneity can also be observed due to transient long-range RNA interactions [166]. Short-range interactions were found to be more conserved than long-range interactions between the virus strains [167]. Different modes of long-range RNA interactions were observed for different viruses [166]. Furthermore, strain-specific structures have also been observed and proposed to influence viral infectivity. For example, the interaction between the 5′UTR and envelope protein-coding region was only found in the pandemic strain of Zika virus [167]. While these RNA structures were found to be essential, their precise functions remain elusive.

### 7.4. Subgenomic Flaviviral RNA (sfRNA)

Small subgenomic flaviviral RNAs (sfRNAs) are formed from the incomplete digestion of the viral genomic RNA at the 3′ UTR by host 5′-3′ exoribonuclease Xrn-1 [154,155,156]. This resistance to degradation by Xrn-1 is conferred by the complex secondary and tertiary structure of the 3′ UTR at PK-1 and PK-2 [154,155,156]. Both PK-1 and PK-2 form a three-way junction to generate a ring-like conformation with the 5′ end of the RNA passing through the centre of the ring [154,155]. In addition to the three-way junction, numerous base pair interactions have been observed to be essential for structure formation [154,155]. This ring-like structure blocks Xrn-1 from accessing the next nucleotide in the 5’ to 3’ direction, enabling the RNA to resist degradation. Although Xrn-1 contains helicase activity, it is unable to bypass the block as it is unable to go through the ring and unwind the structure inside out. However, the ring structure does not inhibit the activity of the viral RNA-dependent RNA polymerase, which acts from the 3′ to 5′ direction, as it enters the ring structure from the outside of the ring. Furthermore, the L3-S4 pseudoknot, observed in the ZIKV xrRNA-1 structure, has been shown to form a continuous stack with the P4 helix [154]. Modelling of Xrn-1 and the xrRNA-1 showed that the P4 helix formed close contacts with the conserved winged-helix region of Xrn-1. This contact may prevent conformational changes in the enzyme, which is important for enzyme activity.

## 8. Future Work

During a flaviviral infection, the virus stages a complete overhaul of the cellular milieu, vastly altering and co-opting innumerous host cell functions. The formation of replication organelles is just one of these alterations, albeit a key alternation, and the wider restructuring of the cell is intrinsically linked to the RC function. To effectively target flaviviral infections, we must obtain a comprehensive understanding of virus’s methods of hijacking cellular processes to create a favourable environment for replication. The other drastic changes to the cell brought about by the virus include the establishment of the network of convoluted membranes, the functional and morphological perturbation of multiple cellular organelles and the reorganisation of the cytoskeleton. The full functional significance of these changes and the mechanisms of their formation remain somewhat elusive, although links to the nonstructural proteins have emerged. As mentioned above, the convoluted membranes are enriched with NS3 and NS2b, and NS4b is thought to elicit structural changes in the mitochondria. Furthermore, NS4a is known to interact with the intermediate filament vimentin, likely as part of the cytoskeletal manipulations [169]. It is probable that the nonstructural proteins have further functions in exploiting hosts that are currently unknown. These processes are a crucial part in the orchestration of efficient and effective virus replication conducted by the nonstructural proteins. Recent developments in focused ion beam cryo-electron microscopy and the improved resolution of subtomogram averaging may allow us to understand the changes that occur on a whole-cell basis throughout different stages of the infection.

As described earlier, the membrane association of the viral replication complex is mediated by NS2A, NS2B, NS4A and NS4B. For the cytoplasmic proteins, NS3 will be anchored to the ER membrane via NS2B while NS5 will be localised to the ER membrane through its interaction with NS3. However, to date, most functional characterisations of viral enzymatic proteins have been carried out in the absence of a membranous system. Through the development of a model membrane system, it can help to limit diffusion and arrange the NS protein on the membrane to trigger the assembly of the replication complex. In doing so, this type of model will be able to provide a more accurate understanding of the intrinsic membrane activity of the viral NS proteins.

Another significant gap in our understanding is the overall structure of the replication complex. Although the structures of key proteins such as NS1, NS3 and NS5 are known, the structures of the other four nonstructural proteins remain elusive, as well as the way they assemble. Additionally, the number of copies of each nonstructural protein within one RC is not completely certain, with some conflicting data on the matter. Furthermore, little is known about the temporal changes to the RC throughout the RNA synthesis process and the life cycle of flaviviruses. These unknowns hamper our understanding of the mechanisms involved in the biogenesis of the complexes, as well as their functioning. Since these processes are pivotal to the replication cycle, they provide key targets in the treatment of flaviviral infections. There is a wealth of literature on the host factors that are subverted by flaviviruses; however, some key questions remain unanswered here. We have yet to fully understand which host factors are incorporated into the complex, and how they contribute to its overall structure. Furthermore, their precise impacts on the processes occurring in the complex and the wider cell are in many cases poorly understood. To effectively design methods of arresting viral replication we must fully comprehend the cumulative effects of these factors. With advancements in electron microscopy, it is now possible to capture multimeric protein complexes that were previously challenging with X-ray crystallography. Using high-resolution cryo-electron tomography, Unchwaniwala et al. revealed the membrane structure and also captured a multimeric Protein A assembly replication complex of the flock house virus [170]. Numerous groups have also reported different multimeric structures of the replication and transcription complex of SARS-CoV-2, from a smaller polymerase complex consisting of nsp7, 8 and nsp12 to an extended complex of nsp7–nsp82–nsp12–nsp132–RNA and nsp9 [171,172,173,174,175]. By doing so, this will help to dissect the molecular details of viral RNA synthesis.

When do the proteins know to transition from negative-sense RNA synthesis, to positive-sense RNA synthesis, to RNA capping? What are the molecular switches? How does NS5 recognise the viral double-stranded replication form and distinguish the negative from the positive strand in the RNA replication form/intermediate? Currently, there is a need for a robust in vitro replication system to segregate various molecular events, particularly the positive-sense RNA synthesis and RNA capping. With a robust system, it will also allow scientists to characterise the molecular architectures of the replication complex during the different stages of replication, and the dynamic changes from one process to another.

## Figures and Tables

**Figure 1 viruses-13-00956-f001:**
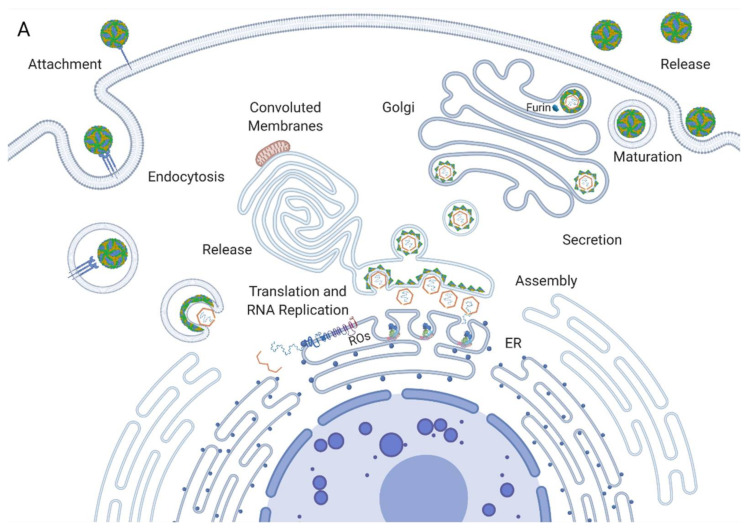
The flavivirus replication cycle. (**A**) Cartoon representation of the steps in the replication cycle, as detailed earlier in the section. (**B**) The translation and RNA replication step of replication are enlarged, showing the viral polyprotein, possible arrangement of the proteins in the complex, and the virion packaging mechanism. These steps require further study to fully elucidate how these structures assemble and function [15,16,17,18,19,20,21]. Created with Biorender.com.

**Figure 2 viruses-13-00956-f002:**
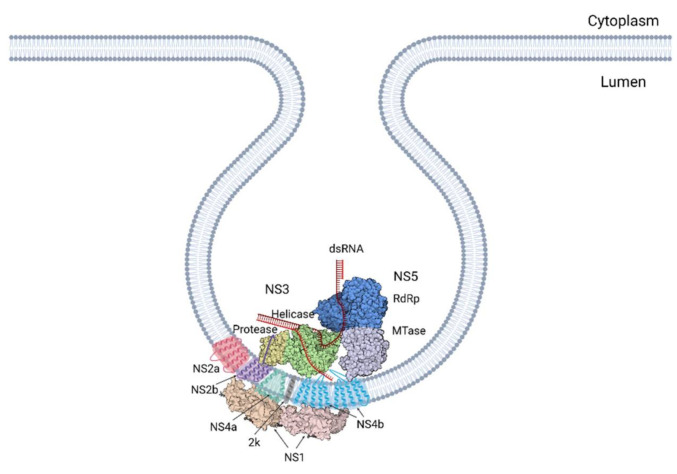
Flavivirus replication organelle. A representation of a possible configuration of the replication complex based on current understanding. Further study is required to elucidate how the interplay between these, the ROs and the host influences viral propagation [15,16,17]. Created with Biorender.com.

**Figure 3 viruses-13-00956-f003:**
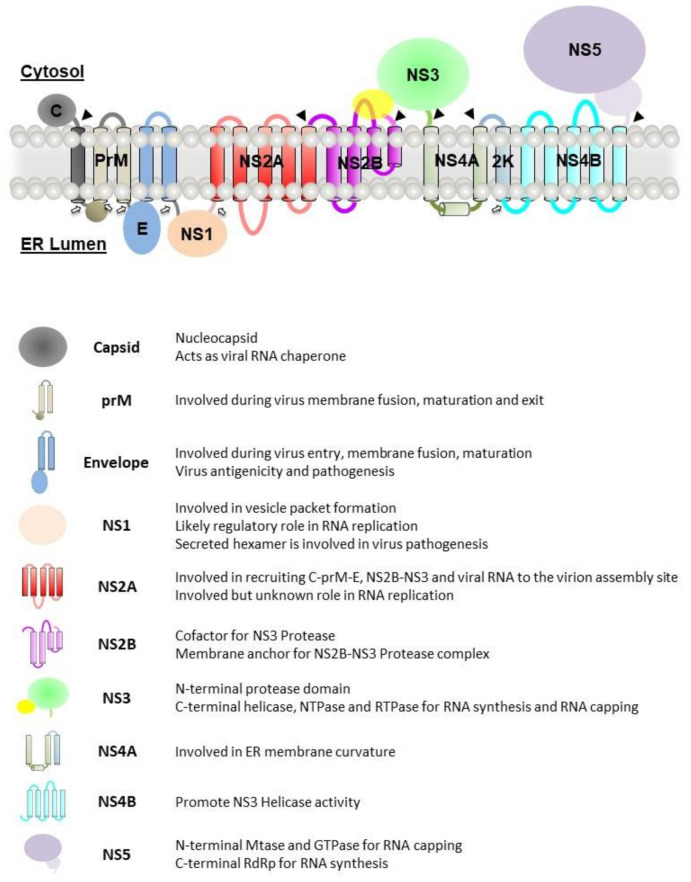
Cartoon representation of the flavivirus polyprotein. The cleavage site of viral protease is represented by a solid arrow, while the cleavage site by host protease is represented by an open arrow. A summary of the function of the viral proteins is described.

**Figure 7 viruses-13-00956-f007:**
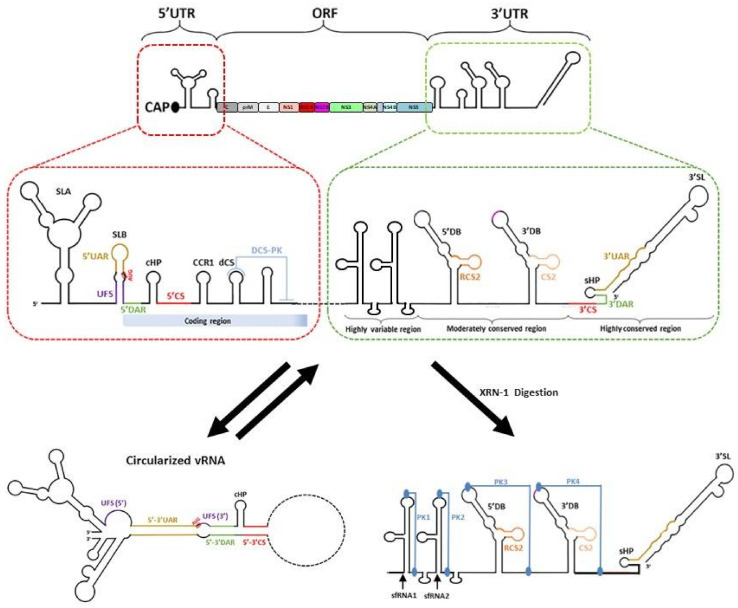
Schematic representation of the flavivirus genome. The figure is adapted from [145]. The viral genome is capped with a single open reading frame that is flanked by both 5′ and 3′ UTRs. Functional RNA elements in the 5′ UTR include stem-loop (SL) A, SLB, the upstream AUG region (UAR), UAR-flanking stem (UFS) and the capsid-coding region hairpin (cHP). The 3′ UTR can be classified into three regions based on sequence conservation and includes functional elements such as DB and 3′SL. These secondary structures and functional RNA elements are shown and annotated. The viral genome is also proposed to undergo genome cyclisation. The circularisation is mediated by sets of inverted complementary sequences 5′UAR-3′UAR, and 5′ downstream AUG region (DAR)-3′DAR, 5′ cyclisation sequence (CS)-3′CS, as illustrated. The predicted subgenomic RNAs sfRNA1 and sfRNA2 are also formed depending on the location in which Xrn-1 is installed.

## Data Availability

Not applicable.

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
