# Peer review of "Molecular Insights into the Flavivirus Replication Complex"

_viruses, 2021, doi:10.3390/v13060956_

Round 1
Reviewer 1 Report
The review by vd Elsen summarizes the current understanding of the flavivirus replication complex, formation of membrane vesicles, and individual non-structural (NS) proteins. The review is well organized, but grammatical mistakes and typos distract the reader from the review. Additionally, the authors may consider the following points.
1. Figures 1-3 are not referenced in the main text. Figure 1B legend is missing. In Figures 1B and 2, the cytoplasmic and lumen sides of ER should be indicated.
2. Table 1 is not formatted correctly. The NS protein figures need a description in the figure legend and the main text.
Author Response
The review by vd Elsen summarizes the current understanding of the flavivirus replication complex, formation of membrane vesicles, and individual non-structural (NS) proteins. The review is well organized, but grammatical mistakes and typos distract the reader from the review. Additionally, the authors may consider the following points.
Thank you for your comments. We have had the manuscript extensively edited for English language and style required by the Journal.
- Figures 1-3 are not referenced in the main text. Figure 1B legend is missing. In Figures 1B and 2, the cytoplasmic and lumen sides of ER should be indicated.
Thank you for your comments. A caption has now been added for this part of the figure and the labels for cytoplasm and lumen have been included.
- Table 1 is not formatted correctly. The NS protein figures need a description in the figure legend and the main text.
Thank you for your comments. Table 1 are being separated into Table 1 – 3 and properly formatted. Additional details were also added to address the differences in the protein structures. The protein figures are being descripted separately in Figure 4 and 6.
Reviewer 2 Report
Title
Molecular Insights into the Flavivirus Replication Complex
Authors
Kaïn van den Elsen , Jun Ping Quek , Dahai Luo
Comments
This is a very nice and complete review on Flavivirus replication. In addition, it describes the NS proteins of the flaviviruses, their interaction and the viral RNA.
In general, the review is well written and easy to understand. Although I am not a native English speaker myself, I noticed small language mistakes here and there. In particular, the conjugation should be checked and an “s” is often missing in the plural form. I would therefore recommend to ask a native English speaker to proofread the review. This would also help to remove the few spelling mistakes (“dsRNS” instead of dsRNA, “invaignations” instead of invaginations, “CRISPR screes” instead of screens,…)
Coming back to the contents of the review, I would like to suggest the following improvements:
1/ Update the statistics on peoples infected/dying from Dengue: the cited publication is from 2013. It is probably possible to find recent numbers from the WHO homepage.
2/ Flavivirus Replication Cycle
It is written that: “The replication complexes replicate the viral RNA through a negative RNA intermediate which is then packaged into new nucleocapsids and envelopes, creating immature virions.”
This sentence is unclear. It sounds as if the negative RNA intermediate is packaged in the virion and not the positive strand transcribed from the negative RNA intermediate.
3/ Figure 1
There is no caption for the “B” part.
4/ Convoluted membranes
Although convoluted membranes are less well described as RO, it would be nice to have a figure showing their configuration like figure 2 is showing the configuration of RO.
5/ NS3
The structure of NS3 is well described but it would be nice to add a figure to illustrate this.
Author Response
Reviewer 2
In general, the review is well written and easy to understand. Although I am not a native English speaker myself, I noticed small language mistakes here and there. In particular, the conjugation should be checked and an “s” is often missing in the plural form. I would therefore recommend to ask a native English speaker to proofread the review. This would also help to remove the few spelling mistakes (“dsRNS” instead of dsRNA, “invaignations” instead of invaginations, “CRISPR screes” instead of screens,…)
Thank you for your comments. We have had the manuscript extensively edited for English language and style required by the Journal.
Coming back to the contents of the review, I would like to suggest the following improvements:
1/ Update the statistics on peoples infected/dying from Dengue: the cited publication is from 2013. It is probably possible to find recent numbers from the WHO homepage.
Thank you for your comments. A sentence has been added which includes the number of cases reported to the WHO up to 2019.
2/ Flavivirus Replication Cycle
It is written that: “The replication complexes replicate the viral RNA through a negative RNA intermediate which is then packaged into new nucleocapsids and envelopes, creating immature virions.”
This sentence is unclear. It sounds as if the negative RNA intermediate is packaged in the virion and not the positive strand transcribed from the negative RNA intermediate.
Thank you for your comments. This has now been amended to specify that it is the positive RNA strand that is packaged into new virions.
3/ Figure 1. There is no caption for the “B” part.
Thank you for your comments. A caption has now been added for this part of the figure.
4/ Convoluted membranes. Although convoluted membranes are less well described as RO, it would be nice to have a figure showing their configuration like figure 2 is showing the configuration of RO.
Thank you for your comments. A representation of the convoluted membranes has been added into the replication cycle.
5/ NS3. The structure of NS3 is well described but it would be nice to add a figure to illustrate this.
Thank you for your comments. Figure 4 is added to describe the structure of NS3.
Reviewer 3 Report
This is an excellent review of current knowledge, and it is well-organized and well-written.
There are many typographical errors, incorrect singular/plural, tense problems, and incorrect article use ('the', 'a'). These are easily corrected with a very careful read-through.
Additional minor comments:
p2.
The Flavivirus Replication Cycle, first paragraph, needs a reference for each sentence, and mention of the figure.
Grammatically, this sentence says neg RNA is packaged into virions:
The replication complexes replicate the viral RNA through a negative RNA intermediate which is then packaged into new nucleocapsids and envelopes, creating immature virions
p3.
Figure caption needs to have a legend that explains the green blobs and blue dots, and panels A and B mentioned.
-ve is not a standard abbreviation.
p8
last line should say 'indicate their distinct roles'.
p9
sentence missing a word:
CRISPR-Cas9 genetic screens have also in to the fore lately as a method to study flaviviral pathogenesis .
p11
Table 1 needs every pdb to have the protein name included in the remarks section, which also needs clarification. For example, how can 5M2X and 5M2Z both be conformation 1 of the same protein? There must be something different, which should be stated
p14
unclear what 2K is - the name of that helix? (line 89/90).
Author Response
Reviewer 3
There are many typographical errors, incorrect singular/plural, tense problems, and incorrect article use ('the', 'a'). These are easily corrected with a very careful read-through.
Thank you for your comments. We have had the manuscript extensively edited for English language and style required by the Journal.
Additional minor comments:
p2. The Flavivirus Replication Cycle, first paragraph, needs a reference for each sentence, and mention of the figure.
Thank you for your comments. These have now all been added.
Grammatically, this sentence says neg RNA is packaged into virions:
The replication complexes replicate the viral RNA through a negative RNA intermediate which is then packaged into new nucleocapsids and envelopes, creating immature virions
Thank you for your comments. This has now been amended to specify that it is the positive RNA strand that is packaged into new virions.
p3. Figure caption needs to have a legend that explains the green blobs and blue dots, and panels A and B mentioned.
-ve is not a standard abbreviation.
Thank you for your comments. This has been replaced with the word negative.
p8. last line should say 'indicate their distinct roles'.
Thank you for your comments. This has now been corrected.
p9. sentence missing a word:
CRISPR-Cas9 genetic screens have also in to the fore lately as a method to study flaviviral pathogenesis .
Thank you for your comments. This has now been amended to ‘come in to the fore’.
p11. Table 1 needs every pdb to have the protein name included in the remarks section, which also needs clarification. For example, how can 5M2X and 5M2Z both be conformation 1 of the same protein? There must be something different, which should be stated
Thank you for your comments. Table 1 are being separated into Table 1 – 3. Additional details were also added to address the differences in the protein structures.
p14. unclear what 2K is - the name of that helix? (line 89/90).
Thank you for your comments. We have rephased the sentence to reflect 2k as the name of the last transmembrane helix.
Round 2
Reviewer 1 Report
The revised manuscript by Elsen et al. includes the suggested changes. It reads very well. I have a minor suggestion. For completeness, the authors could include the NS5 structure, 5CCV in Table 3.
Author Response
We thank the reviewer to point out the overlooked DENV3 NS5 structure (pdb code 5CCV) and the associated publication (Klema, V.J. PLoS Pathog 12: e1005451-e1005451). We have now incorporated the structure into the table 3 and cited the paper.